# The dual burden of animal and human zoonoses: A systematic review

Liz P. Noguera Z. [1,2☯]*, Duriya Charypkhan[1,2☯], Sonja Hartnack[1], Paul R. Torgerson[1], Simon R. Rüegg[1]

**1** Section of Epidemiology, Vetsuisse Faculty, University of Zürich, Zürich, Switzerland, **2** Epidemiology and Biostatistics, Life Science Zurich Graduate School, University of Zurich, Zurich, Switzerland

☯ These authors contributed equally to this work.
* lizpaola.noguerazayas@uzh.ch, lpnogal@gmail.com

**Data Availability Statement:** All relevant data are within the manuscript and its Supporting Information files. Scripts are available at https://github.com/LizPNZ/Dual-burden-of-zoonosis.

## Abstract

### Background

Zoonoses can cause a substantial burden on both human and animal health. Globally, estimates of the dual (human and animal) burden of zoonoses are scarce. Therefore, this study aims to quantify the dual burden of zoonoses using a comparable metric, "zoonosis Disability Adjusted Life Years" (zDALY).

### Methodology

We systematically reviewed studies that quantify in the same article zoonoses in animals, through monetary losses, and in humans in terms of Disability Adjusted Life Years (DALYs). We searched EMBASE, Web of Science, Scopus, PubMed, and Google Scholar. We excluded articles that did not provide the data to estimate the zDALY or those for which full text was not available. This study was registered at PROSPERO, CRD42022313081.

### Principal findings/Significance

We identified 512 potentially eligible records. After deduplication and screening of the title and abstract, 23 records were assessed for full-text review. Fourteen studies were included in this systematic review. The data contains estimates from 10 countries, a study at continental level (Asia and Africa), and 2 studies on a global scale.

Rabies was the most frequently reported zoonosis where zDALYs were calculated, based on the following included studies: for Kazakhstan 457 (95% CI 342–597), Viet Nam 5316 (95% CI 4382–6244), Asia 1,145,287 (90% CI 388,592–1,902,310), Africa 837,158 (90% CI 283,087–1,388,963), and worldwide rabies 5,920,014 (95% CI 1,547,860–10,290,815). This was followed by echinococcosis, the zDALYs in Peru were 2238 (95% CI 1931–2546), in China 1490 (95% CI 1442–1537), and worldwide cystic echinococcosis 5,935,463 (95% CI 4,497,316–7,377,636). Then, the zDALYs on cysticercosis for Mozambique were 2075 (95% CI 1476–2809), Cameroon 59,540 (95% CR 16,896–101,803), and Tanzania 34,455 (95% CI 12,993–76,193). Brucellosis in Kazakhstan were 2443 zDALYs (95% CI 2391–2496), and brucellosis and anthrax in Turkey 3538 zDALYs (95% CI 2567–

**Funding:** The study was partially funded by "BECAL" (https://www.becal.gov.py/) 7th/2019 – Grant recipient Liz P. Noguera Z. The funders had no role in study design, data collection and analysis, decision to publish, or preparation of the manuscript.

**Competing interests:** The authors have declared that no competing interests exist.

6706). Finally, zDALYs on leptospirosis in New Zealand were 196, and Q fever in Netherlands 2843 (95% CI 1071–4603).

The animal burden was superior to the human burden in the following studies: worldwide cystic echinococcosis (83%), brucellosis in Kazakhstan (71%), leptospirosis in New Zealand (91%), and brucellosis, and anthrax in Turkey (52%). Countries priorities on zoonoses can change if animal populations are taken into consideration.

## Author summary

Zoonoses impact humans and animals in several ways. Unfortunately, the burden of zoonoses is usually not characterized and quantified through integrated human and animal metrics. Our study is the first systematic review to assess the dual burden of zoonotic diseases in humans and animals globally. In the considered set of human and animal burden of zoonoses, the zDALY due to animal disease varied from 0.005% to 91%. Therefore, metrics encompassing both burdens are likely to change decision-making regarding the prevention and control of zoonoses. Implementing a "One Health" approach will require the application of such metrics. We believe that quantification of the dual burden of the diseases is a key to improving zoonosis prioritization decision-making, and resource allocation. This study outlines the need for integrated studies on zoonoses and reporting of data with a comparable metric.

## Introduction

Zoonoses are diseases that can be transmitted directly or indirectly from animals to humans (and vice versa, hence anthroponoses). Around 6 in 10 human infections are zoonotic [1]. In the human population, early detection of zoonoses prevents loss of life, well-being, money, time, and productivity. By definition, zoonoses harm domestic animals and may threaten wildlife [2]. Zoonotic diseases also incur financial costs, including those caused by losses to humans, animals, and the environment. Integrated surveillance in animals can provide significant benefits, including knowledge generation. The additional economic benefit of zoonoses surveillance might help decide how much data integration is sought, impacting surveillance types, diseases, and geographical settings. Recent pandemics have highlighted the need for surveillance systems for zoonotic events, and the need for better communication across the human-animal-ecosystems continuum [3]. Because human, animal, and ecosystem health are intimately related, surveillance should be organized in an integrated way [4]. This allows for a comprehensive risk assessment and the design of appropriate responses [5].

The business case for a "One Health" (OH) approach to mitigation of zoonoses has been presented as a framework [6] which includes the creation of one health surveillance and response programs for future emerging diseases. Animal health surveillance data can be used to inform public health messaging, control measures along the food chain, and establish public health surveillance if a pathogen is present in the human population and public health action is required.

In general, the impact of zoonotic diseases on the human population is measured by financial cost, mortality, morbidity, or other indicators known as disease burden [7]. The specific burden of disease on humans can be quantified using the Disability Adjusted Life Years (DALY) [8]. The DALYs consist of the loss of health due to a disease (or disability) and

premature mortality [9]. Methods that estimate the human disease burden in monetary terms include costs associated with the diagnostics and treatment of the disease, the statistical value of a human life, costs related to the loss of productivity or loss of income in humans.

The direct impact of animal disease is studied using various economic models. For example, the burden of diseases can be quantified through the money spent on the disease intervention programs, or money accounted for the loss of animal productivity (less milk/meat yield, etc.). The challenge of economic analysis in a OH context is that the boundaries of the system for which costs and benefits incur can be extended or restricted arbitrarily and hence alternative economic models are needed.

A pragmatic approach to consider the combined burden on human and animal health has been proposed as "*zoonosis Disability Adjusted Life Years*" (zDALYs) [10]. The zDALYs extends the DALY framework to domestic animals. The idea behind this indicator is that the animal burden estimated as monetary losses can be converted to Animal Loss Equivalents (ALE). The ALE is basically a metric that reflects the time trade-off for human life years to "replace" the animal loss, e.g., it is the amount of time that a farmer would need to spend to recover the losses.

Despite the availability of data on the zoonosis burden in humans and animals regarding monetary and societal costs separately, only a few studies have estimated the dual burden in animals and humans [11–13]. We conducted a systematic review of the literature focusing on socio-economic burden of zoonoses worldwide and estimated the zDALYs of such studies.

## Methods

### Search strategy and selection criteria

We followed the guidelines for "Preferred Reporting Items for Systematic reviews and Meta-Analyses [14]. A medical librarian assisted in the development of the search syntax.

We searched electronic academic databases (Embase, Ovid Medline, Scopus, Web of Science) and internet search engines (Google Scholar) for observational epidemiological studies on, at least, a zoonotic disease that includes human disease burden in DALYs and animal disease burden expressed in monetary terms. We included all peer-reviewed studies from an unrestricted period until November 2021. We excluded non-observational epidemiological studies such as experimental studies (for example, only molecular biology studies), clinical cases, scientific correspondence, or mathematical models without data on the burden of zoonoses. The data sources and search terms with results are provided in the S1 Table.

### Data extraction

According to the eligibility criteria stated above, the identified titles and abstracts were independently reviewed by two reviewers (LPNZ and DC). Then, DC and LPNZ independently assessed the full texts of the included papers and documented the reasons for exclusions. The eligibility disagreements were resolved by group discussion.

The data were independently extracted, and double entered into a Microsoft Excel spreadsheet by the two reviewers. For each study, the size of human and animal populations, diseases, DALYs, and associated animal losses were extracted.

### Data analysis

We used the DALYs and animal loss reported in previous studies. We fitted the data according to what previous authors described in the methodology and results in order to simulate the data distribution, and uncertainty. For this, we used the reported lower and upper bounds.

Based on the data available, we estimated the Animal Loss Equivalents (ALE) of each finding to calculate the zoonosis Disability Adjusted Life Years (zDALY). We divided the annual monetary value of animal health losses by the Gross National Income (GNI) per capita in US$ at the period of the study. The GNIs were obtained from World Bank Open Data. For the economic losses that were in a different currency than the US$, we converted it into the US$ at the year of the study using a historical currency converter [15].

ALE = annual monetary value of animal health losses/GNI per capita in US$ at the period of the study

We computed the zDALY, adding the DALY of the findings to the ALE that we estimated.

$$zDALY = DALY + ALE$$

To account for the uncertainty of all estimates, we generated random numbers between the lower and upper bounds of the distributions from the previous studies. We set 100,000 iterations for each estimation. According to the original studies, we reported the 50, 2.5, and 97.5 percentiles of the estimates, and 50, 5, 95 percentiles. We have also kept the terms that previous studies used to express uncertainty (e.g., confidence Interval, confidence region, prediction interval).

We performed the analyses in R 4.1.3. Scripts are available at https://github.com/LizPNZ/Dual-burden-of-zoonosis.

We estimated ALEs and zDALYs for each study with available data over the study period. We reported bias qualitatively through the ROBIS tool [16]. The ROBIS tool encompasses three phases, the first being optional, as it assesses the relevance of the review and the target question. We considered Phase 1 redundant because its questions are a repetition of the inclusion criteria already described in the protocol and methodology. Phase 2 includes the identification of concerns with the review process, and Phase 3, the judgment of risk of bias.

This study is registered at **PROSPERO, CRD42022313081**, and its protocol is in a preprint form [17].

## Results

We identified 552 articles through electronic database searches (Fig 1). After removing 140 duplicates, 412 articles were screened for titles and abstracts. The full texts of 23 articles were reviewed and 9 were excluded at this stage. Thus, 14 articles are included in this review (Table 1, S1 Text). Common reasons for exclusion at the full-text screening stage were no relevant data or the absence of data on animal monetary losses, DALYs in humans, or absence of full-text. The list of articles excluded at the full-text stage with the brief reasons for exclusion can be found in S2 Table.

Publications on zoonoses considering human and animal populations that met the inclusion criteria started in 2005. Most reported zoonoses were parasitic, whereas no fungal zoonosis was reported (S1 Fig). The most frequently reported zoonoses were rabies, and food-borne diseases such as cystic echinococcosis, and cysticercosis.

The studies considered mainly low- and middle-income countries, except for the Netherlands and New Zealand. Only two studies on rabies and cystic echinococcosis were on a global scale, and one study on rabies in two continents: Africa, and Asia (Fig 2). The preferred currency to measure the economic loss was the U.S. dollar for 12 articles, and the euro for studies in Cameroon and the Netherlands.

All studies performed their assessment of the monetary impact of the disease. In humans, it comprised the costs associated with direct treatment of the medical condition and indirect

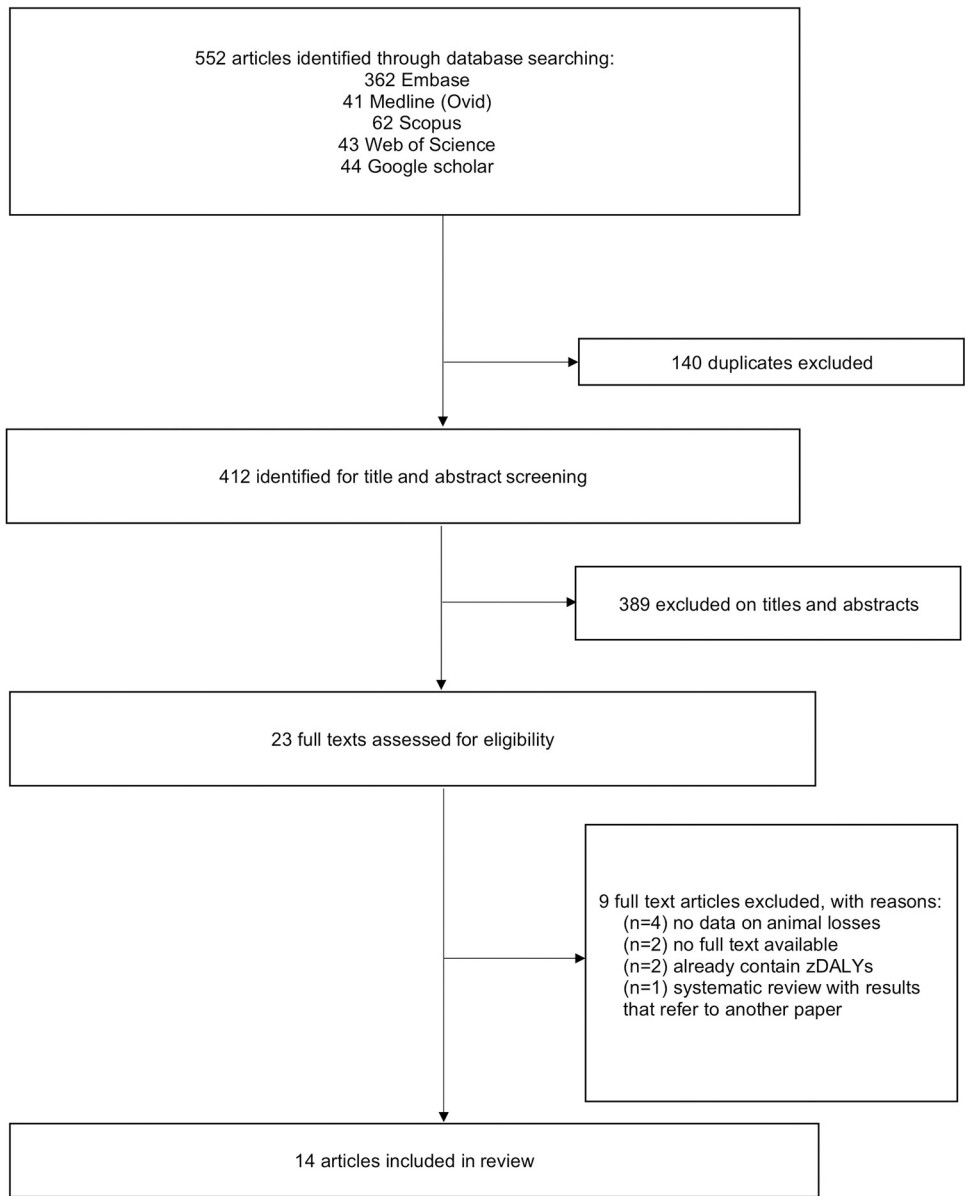

**Fig 1. Literature search and article inclusion.**

costs associated with for example, transportation. In animals, it was costs associated with lost productivity, organ condemnation, or death.

Ten articles used stochastic methods for their estimations, expressing their uncertainty in a 95% Confidence Interval (CI), Uncertainty Interval (UI), Confidence Region (CR), Prediction Interval (PI), and one with a 90% CI (Table 2).

Four papers estimated the burden of rabies: Africa and Asia, Viet Nam, Kazakhstan, and worldwide. The countries included in the worldwide study on rabies, Africa and Asia are listed in the S3 Table. Viet Nam reported the DALYs by age (26, 31, 36). Whereas Kazakhstan reported the values on rabies without post-exposure prophylaxis (PEP). The total zDALYs per capita was higher in Africa (11 zDALYs per 10,000 population) than Asia (3 zDALYs per 10,000 population).

**Table 1. Findings in the dual burden of zoonoses (ordered by ascending year of the data source).**

| Authors | Period of data source | Zoonotic disease/ pathogen | Country/ Region | DALY | Uncertainty | Animal species | Animal loss |
|---|---|---|---|---|---|---|---|
| Knobel et al. [18] | Human data: 1996–2000, 2003 Livestock cost: 2002 | Rabies | Africa and Asia | **Africa:** 747,918 (217,954–1,449,114); **Asia:** 1,039,119 (302,324–1,983,646) **Total without PEP:** 9,504,237 (4,848,684–15,264,050) **Total:** 1,787,886 (799,615–2,984,109) | 90% CI | Livestock | **Africa:** US$ 1.7 (1.5–1.9) **Asia:** US$ 10.5 (9.4–11.8) **Total:** US$ 12.3 (11–13.7) (All values in million dollars) |
| Budke et al. [19] | 1996–2003 | Cystic echinococcosis | Worldwide | **Unadjusted:** 285,407 (218,515–366,133) **Adjusted for underreporting:** 1,009,662 (862,119–1,175,654) | 95% CI | Livestock | **Unadjusted:** US$ 1,249,866,660 (942,356,157–1,622,045,957) **Adjusted for underreporting:** US$ 2,190,132,464 (1,572,373,055–2,951,409,989) |
| Budke et al. [20] | Human data: 2001–2003 Animal data: 1980, 1997 | Echinococcosis | China (Shiqu County) | 1100 | 95% CI (for animal loss estimation) | Livestock (calves, yaks, meat) | **Total losses (excluding losses in calf production, carcass weight, and yak hide):** US$ 278,292 (240,829–318,249) **Total losses (including losses in calf production, carcass weight, and yak hide):** US$ 439,734 (384,342–498,447) |
| Trevisan et al. [21] | 2007 | Cysticercosis (*Taenia solium*) | Mozambique (Angónia district) | 2003 (1433–2762) | 95% UI | Pigs | US$ 22,282 (12,315–35,647) |
| Praet et al. [22] | 2008 | Cysticercosis (*Taenia solium*) | Cameroon | 45,838 (14,108–103,469) | 95% CR | Pigs | € 478,844 (369,587–601,325) |
| Moro et al. [23] | 2010 | Cystic echinococcosis | Peru | 1,139 (861–1,489) | 95% CI | Livestock | US$ 3,846,754 (2,676,181–4,911,383) |
| Hampson et al.[24] | 2010 | Rabies | Worldwide Asia 2 Asia 3 Asia 4 China India Indonesia North Africa Congo Basin West Africa SADC Andean Brazil Caribbean Central America Southern Cone Eastern Europe Eurasia Middle East | 3,714,333 (1,316,000–10,519,000) 357,015 (80,000–655,000) 160,801 (75,000–853,000) 16,521 (10,000–83,000) 374,851 (60,000–674,000) 1,301,865 (377,000–3,436,000) 12,311 (12,000–198,000) 123,074 (38,000–467,000) 449,382 (244,000–1,031,000) 375,023 (206,000–971,000) 398,164 (157,000–1,713,000) 1,582 (0–4000) 1,023 (0–2000) 8,581 (4000–17,000) 495 (0–3000) 270 (0–1000) 1,948 (0–5000) 117,116 (46,000–368,000) 14,310 (6000–39,000) | 95% CI | Livestock | **Total:** 129.55 **Asia 2:** 2.073 **Asia 3:** 0.564 **Asia 4:** 11.248 **China:** 4.235 **India:** 9.050 **Indonesia:** 6.384 **North Africa:** 2.756 **Congo Basin:** 0.481 **West Africa:** 6.684 **SADC:** 4.600 **Andean:** 10.753 **Brazil:** 16.620 **Caribbean:** 2.575 **Central America:** 31.308 **Southern Cone:** 4.710 **Eastern Europe:** 10.460 **Eurasia:** 4.451 **Middle East:** 0.592 (In thousands of US$) |

(*Continued*)

**Table 1.** (Continued)

| Authors | Period of data source | Zoonotic disease/pathogen | Country/Region | DALY | Uncertainty | Animal species | Animal loss |
|---|---|---|---|---|---|---|---|
| van Asseldonk et al.[25] | 2007–2011 | Q fever | Netherlands | 2462 | ---- | Goats | **Loss culling milk goat:** € 300 /case **Loss breeding prohibition:** € 250/ goat **Total:** € 0.03 Million |
| Trevisan et al. [26] | 2012 | Cysticercosis (*Taenia solium*) | Tanzania | 31,863 (9136–72,078) | 95% UI | Pigs | US$ 2,800,000 (1,100,000–5,400,000) |
| Shwiff et al. [27] | 2005–2014 | Rabies | Viet Nam | 12,339 | ---- | Livestock | US$ 10,344,223 |
| Sultanov et al. [28] | 2003–2015 | Rabies | Kazakhstan | **Total:** 454 (339–593) Without **PEP:** 7827 (4746–12,074) | 95% CI | Livestock (cattle, sheep, horses and camels) | US$ 5,400,000 (4,000,000–7,100,000) |
| Charypkhan et al.[29] | 2006–2015 | Brucellosis | Kazakhstan | 713 | ---- | Cattle, sheep | US$ 21,316,800 |
| Sanhueza et al.[30] | 2013–2019 | Leptospirosis | New Zealand | **At risk of leptospirosis:** 14.07 (1.86–80.73) **Not at risk of leptospirosis:** 3.69 (0.49–21.20) **Total:** 17.76 (2.35–101.93) | 95% PI | Beef cattle, sheep and deer. | US$ 7.92 (3.75–15.48) million |
| Ari et al.[31] | 2016–2018 | Brucella, Anthrax, Tularemia, CCHF, Rabies, Cystic Echinococcosis, Toxoplasmosis | Turkey | **Total:** 1782 **Brucella:** 1068 **Anthrax:** 50 **Tularemia:** 1 **CCHF:** 505 **Rabies:** 113 **Cystic Echinococcosis:** 24 **Toxoplasmosis:** 21 | ---- | Livestock (large and small ruminants) | **Total loss in 2016:** US$ 213,674,967 **Total loss in 2017:** US$ 263,105,316 **Total loss in 2018:** US$ 336,313,908 **Mean of total loss:** US$ 271,031,397 |

Asia 2: Cambodia, Myanmar, Laos, Viet Nam, and Democratic People's Republic of Korea; Asia 3: Bhutan, Nepal, Bangladesh, Pakistan (Himalayan region); Asia 4: Philippines, Sri Lanka, Thailand; SADC: countries in the Southern African Development Community; Eurasia: Afghanistan, Kazakhstan, Kyrgyzstan, Mongolia, the Russian Federation, Turkmenistan, Tajikistan, and Uzbekistan. More information in the S3 Table.

CI: Confidence Interval, UI: Uncertainty Interval, CR: Confidence Region, PI: Prediction Interval

PEP: post-exposure prophylaxis

Cystic echinococcosis *(E. granulosus)* was reported in Peru, Turkey, and on a global scale. In addition, a study in Shiqu County, China, studied both cystic echinococcosis, and alveolar echinococcosis (*E. multilocularis*).

For brucellosis, the Kazakh study only accounted for losses due to slaughtering of the animals and subsequent compensation. Whereas the Turkish study also considered reduced productivity. Besides, the Turkish study was the only one that included bacterial, parasitic, and viral zoonoses. However, we only determined the ALE for brucellosis and anthrax since the animal loss was only available for those diseases. We calculated the total zDALY for all the diseases included in this study.

Since the studies that already estimated zDALYs did not meet the inclusion criteria, we added their findings in the S4 Table.

### Bias assessment–ROBIS

The full ROBIS assessment is provided in the S5 Table. Overall, the risk of bias for this study is low. According to the signaling questions, there were no concerns regarding all the domains

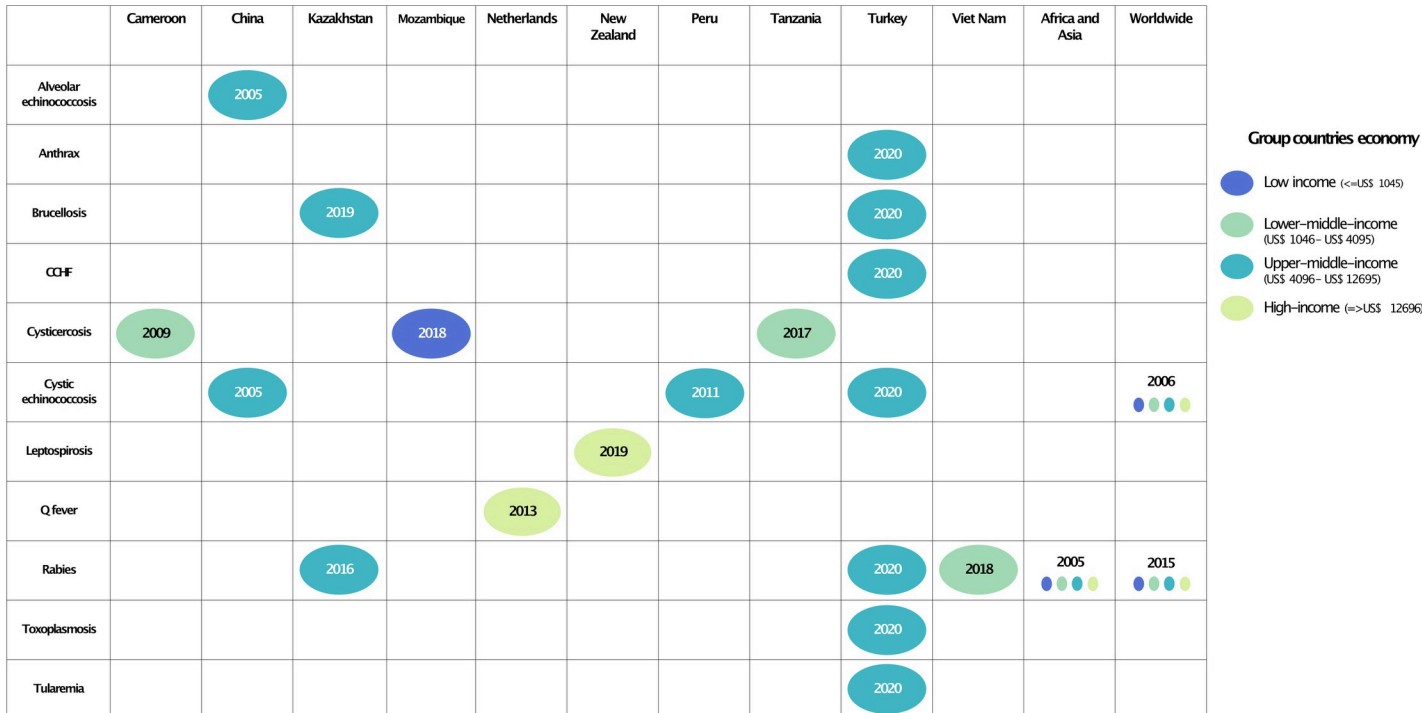

**Fig 2. Zoonoses studied in humans and animals with their year of publication by income countries.**

(study eligibility criteria, identification, selection of studies, and data collection). Therefore, the review is likely to include a high proportion of relevant studies.

However, the last domain (synthesis and findings) outlines that no meta-analysis was performed. We report the reasons in the discussion.

The PRISMA checklist is provided in the S6 Table.

## Discussion

We report the first systematic review that estimates the dual burden of zoonoses in humans and domestic animals based on studies available worldwide. Such information is needed for zoonosis prioritization, and resource allocation since interventions to control zoonoses are frequently carried out in animal hosts. Zoonoses impact health and socio-economic factors in multiple ways, increasing inequity between populations. Zoonoses in low-income countries (LICs) are often under-reported compared to non-zoonotic diseases [32].

Despite the substantial burden caused by zoonoses in humans and animals, the number of studies combining both burdens is scarce. Besides, the use of old data does not reflect the current situation that depicts the dual burden of zoonoses. Studies that include human and animal data for zoonoses are relatively new (published in the last 20 years, S2 Fig). We observed an increased number of reports on the dual burden of diseases over the years. Up to date, only three studies have reported zDALYs: on cystic echinococcosis in Morocco [13], 25 zoonoses in Paraguay [11], *Taenia solium* in Lao PDR [12]. We excluded them from our synthesis since they already contain zDALY values.

The dual burden of zoonoses was reported the most in Asia and Africa. The majority of zoonoses were based on estimations, due to the lack of reports, access to health care, and tools for disease diagnoses. The data source of the global estimates on rabies (Hampson et al.) [33]

**Table 2. Estimates of the dual burden of zoonoses.**

| Zoonotic disease/ pathogen | Based on | Year | Country/ Region | DALY | ALE | zDALY | Uncertainty and distribution |
|---|---|---|---|---|---|---|---|
| Rabies (Lyssavirus) | Knobel et al. [18] | Human data: 1996–2000, 2003 Livestock cost: 2002 | Africa and Asia | **Africa:** 835,380 (281,198–1,387,050) **Asia:** 1,141,077 (3,844,311–1,898,325) **Total:** 1,882,387 (907,507–2,874,205) **Total without PEP:** 10,068,537 (5,373,433–14,747,882) | **Africa:** 1858 (1661–2055) **Asia:** 4157 (3733–4580) **Total:** 7334 (6612–8055) | **Africa:** 837,158 (283,087–1,388,963) **Asia:** 1,145,287 (388,592–1,902,310) **Total:** 1,889,928 (914,795–2,881,607) **Total without PEP:** 10,075,831 (5,380,459–14,755,386) | 90% CI Uniform distribution |
| | Hampson et al.[24] | 2010 | Worldwide | **Asia 2:** 368,376 (94,862–640,037) **Asia 3:** 462097 (94,090–833,514) **Asia 4:** 46619 (11,803–81,145) **China:** 365023 (74,959–658,747) **India:** 1,909,088 (453,985–3,358,527) **Indonesia:** 105605 (16575–193418) **North Africa:** 251,128 (48,721–455,977) **Congo Basin:** 636,550 (263,527–1,011,627) **West Africa:** 587,499 (224,634–952,020) **SADC:** 939,689 (197,503–1,673,558) **Andean:** 1994 (101–3898) **Brazil:** 998 (50–1949) **Caribbean:** 10459 (4308–16,672) **Central America:** 1493 (75–2925) **Southern Cone:** 503 (24–976) **Eastern Europe:** 2497 (128–4875) **Eurasia:** 206583 (54,047–359,951) **Middle East:** 22,594 (6822–38,167) **Total:** 5,916,890 (1,544,600–10,282,026) | **Asia 2:** 420 (44–1611) **Asia 3:** 87 (0.6–453) **Asia 4:** 34 (6–207) **China:** 1448 (405–2477) **India:** 4580 (1439–7724) **Indonesia:** 22 (0–506) **North Africa:** 8 (0.5–73) **Congo Basin:** 3 (0.3–36) **West Africa:** 11 (0–186) **SADC:** 5 (0–57) **Andean:** 2 (0.2–11) **Brazil:** 3 (2–5) **Caribbean:** 0 (0–2) **Central:** 0.03 (0–5) **Southern Cone:** 0 (0–4) **Eastern Europe:** 0.12 (0–2) **Eurasia:** 5 (1–62) **Middle East:** 0.15 (0.02–3) **Total:** 279 (101–466) | **Asia 2:** 367,849 (94,900–641,049) **Asia 3:** 464,757 (94,279–833,473) **Asia 4:** 46485 (11,854–81,205) **China:** 368,536 (76,900–660,044) **India:** 1907787 (457,488–3,364,968) **Indonesia:** 105,310 (16,715–193,698) **North Africa:** 253,229 (48,634–456,088) **Congo Basin:** 638,791 (263,413–1,011,283) **West Africa:** 587,641 (225,199–952,027) **SADC:** 934,682 (196,022–1,674,590) **Andean:** 2009 (104–3905) **Brazil:** 1006 (52–1952) **Caribbean:** 10,467 (4324–16,675) **Central America:** 1491 (74–2925) **Southern Cone:** 500 (26–975) **Eastern Europe:** 2509 (126–4874) **Eurasia:** 206,690 (54,015–360,086) **Middle East:** 22,532 (6848–38,182) **Total:** 5,920,014 (1,547,860–10,290,815) | 95% CI Uniform distribution, Poisson |

*(Continued)*

**Table 2.** (Continued)

| Zoonotic disease/ pathogen | Based on | Year | Country/ Region | DALY | ALE | zDALY | Uncertainty and distribution |
|---|---|---|---|---|---|---|---|
| Rabies (*Lyssavirus*) | Shwiff et al. [27] | 2005–2014 | Viet Nam | **Age 26:** 4956 (3432–6471); **Age 31:** 4450 (3086–5824); **Age 36:** 3955 (2744–5176) | 3985 (1485–6491) | **Age 26:** 5815 (4292–7331); **Age 31:** 5309 (3946–6683); **Age 36:** 4814 (3603–6035) **Total:** 5316 (4382–6244) | 95% CI Uniform distribution |
| | Sultanov et al.[28] | 2003–2015 Human data: 2007, 2010–2015 | Kazakhstan | **Total:** 454 (339–593) **Without PEP:** 7827 (4746–12074) | **Cattle:** 3 (2.8–3.25) **Sheep:** 0.09 (0.07–0.11); **Camel:** 0.016 (0.009–0.03) **Horse:** 0.3 (0.24–0.42) **Total:** 3.42 (3.16–3.7) | Cattle: 457 (342–596) Sheep: 454 (339–594) Camel: 454 (339–594) Horse: 339 (454–594) **Total:** 457 (342–597). **Without PEP:** Cattle: 7830 (4749–12,077) Sheep: 7827 (4746–12,074) Camel: 7827 (4746–12,074) Horse: 7827 (4746–12,076) **Total:** 7831 (4749–12,077) | 95% CI Gamma distribution |
| Cystic echinococcosis (*E. granulosus*) | Budke et al. [19] | 1996–2003 | Worldwide | **Unadjusted:** 292,111 (222,377–362,385) **Adjusted for underreporting:** 1,019,530 (869,875–1,167,877) | **Unadjusted:** 2,782,397 (2,084,548–3,489,591) **Adjusted for underreporting:** 4,916,173(3,495,999–6,341,741) | **Unadjusted:** 3,075,118 (2,371,693–3,788,135) **Adjusted for underreporting:** 5,935,463 (4,497,316–7,377,636) | 95% CI Uniform distribution |
| | Moro et al. [22] | 2010 | Peru | 1139 | 1099 (792–1407) | 2238 (1931–2546) | 95% CI Uniform distribution |
| Echinococcosiss: alveolar echinococcosis (*E. multilocularis*) and cystic echinococcosis (*E. granulosus*) | Budke et al. [20] | 2001–2003 | China (Shiqu County) | 1100 | **Total losses** (excluding losses in calf production, carcass weight, and yak hide): 247 (214–279) **Total losses** (including losses in calf production, carcass weight, and yak hide): 389 (342–438) | **Total losses** (excluding losses in calf production, carcass weight, and yak hide): 1347 (1314–1379) **Total losses** (including losses in calf production, carcass weight, and yak hide): 1490 (1442–1537) | 95% CI Uniform distribution |
| Cysticercosis (*Taenia solium*) | Trevisan et al.[21] | 2007 | Mozambique (Angónia district) | 2027 (1428–2761) | Without the proportion of pigs sold: 141 (81–230) **Total:** 47 (27–76) | Without the proportion of pigs sold: 2173 (1569–2909) **Total:** 2075 (1476–2809) | 95% UI Gamma distribution |
| | Praet et al. [22] | 2008 | Cameroon | 58,987 (16,329–101,231) | 568 (439–697) | 59,540 (16,896–101,803) | 95% CR Uniform distribution |
| Cysticercosis (*Taenia solium*) | Trevisan et al.[26] | 2012 | Tanzania | 30,443 (9264–72,115) | 3985 (1485–6491) | 34,455 (12,993–76,193) | 95% UI Gamma distribution; Uniform distribution |

(*Continued*)

**Table 2.** (Continued)

| Zoonotic disease/ pathogen | Based on | Year | Country/ Region | DALY | ALE | zDALY | Uncertainty and distribution |
|---|---|---|---|---|---|---|---|
| Brucellosis (*Brucella spp*) | Charypkhan et al.[29] | 2006–2015 | Kazakhstan | 713 (661–766) | 1730 (1729–1731) | 2443 (2391–2496) | 95% CI Poisson distribution |
| Brucella, Anthrax, Tularemia, CCHF, Rabies, Cystic Echinococcosis, Toxoplasmosis | Ari et al.[31] | 2016–2018 | Turkey | Brucella: 1083 (818–1314) Anthrax: 30 (0–135) **Total** (Brucella, Anthrax, Tularemia, CCHF, Rabies, Cystic Echinococcosis, Toxoplasmosis): 1686 (1463–2207) | Brucella large ruminant: 1410 (840–3324) Brucella small ruminant: 265 (119–831) Brucella total: 1675 (959–4155) Anthrax large ruminant: 116 (97–240) Anthrax small ruminant: 56 (46–111) Anthrax total: 3176 (1103–7456) **Total:** 1851 (1104–4500) | Brucella large ruminant: 2493 (1659–4637) Brucella small ruminant: 1348 (937–2144) Brucella total: 2758 (1778–5467) Anthrax large ruminant: 127 (116–375) Anthrax small ruminant: 76 (56–246) Anthrax total: 173 (166–486) **Total:** 3538 (2567–6706) | 95% CI Poisson distribution |
| Q fever (*Coxiella burnetti*) | van Asseldonk et al.[25] | 2007–2011 | Netherlands | 2833 (1071–4603) | 2.86 (1.07–4.6) | 2843 (1071–4603) | 95% CI Uniform distribution |
| Leptospirosis (*Leptospira spp*) | Sanhueza et al.[30] | 2013–2019 | New Zealand | **At risk of leptospirosis:** 14.07 (95% PI: 1.86–80.73) **Not at risk of leptospirosis:** 3.69 (95% PI: 0.49–21.20) **Total:** 17.76 (95% PI: 2.35–101.93) | 178 | **At risk of leptospirosis:** 192 **Not at risk of leptospirosis:** 182 **Total:** 196 | ---- |

The sum of values may not be exact since they are based on estimations randomly generated. Most values are rounded to two significant figures.

and the one reported in Asia and Africa (Knobel et al.) [18] have seven years difference. Both studies applied different ranges of uncertainty to their estimates and used different clusters. Therefore, comparing the zDALYs from Asia and Africa in both studies is slightly difficult. We report higher zDALYs for estimates from Hampson's study. If post-exposure prophylaxis is not considered, the burden increased by 5 times, because rabies is lethal, and hence the high DALYs contribute to higher zDALYs. Comparing the global rabies estimates provided by the Global Burden of Diseases (GBD) [34], and Hampson et al., the median of the latter was 2,665,145 DALYs more than the GBD's in 2010 (the year of the data source of Hampson et al. study.) However, the GBD estimated 2,529,389,250 DALYs more than Hampson's estimation for rabies in 2015 (year of publication of Hampson's study.)

Among diseases included in this review, echinococcosis was the most reported parasitic zoonosis. Cystic echinococcosis being the most common form reported. Echinococcosis causes a considerable burden because its treatment is expensive and complicated [35]. Alveolar echinococcosis (*E. multilocularis*) is considered rare worldwide, except for China, Russia, and the Kyrgyz Republic [36,37]. Alveolar echinococcosis (AE) rarely affects agricultural animals or pets (except for exceedingly rare cases of AE in dogs when they act as an intermediate host), so the health burden on animals is negligible. Dogs are common definitive hosts but do not show any clinical symptoms. Cystic echinococcosis on a global scale was the only disease that

had higher ALE compared to the DALY. Therefore, the animal burden had more influence on the total zDALYs of cystic echinococcosis worldwide. For the global estimation of cystic echinococcosis, *Budke et al.* presented it as adjusted and unadjusted DALYs. They were higher than GBD's without exceptions (including period of data source and publication). The least difference was between the unadjusted values and GBD, mainly in 1996. For that year, the difference was 106,017 DALYs (with unadjusted values) and 833,436 DALYs (adjusted values). The unadjusted DALYs were similar to but higher than 285,000 DALY estimates for CE by the Foodborne Disease Burden Epidemiology Reference Group (FERG)– 184,000 DALYs [38]. This difference may be due to the lower disability weight (DW) used by FERG and GBD (abdominal discomfort) compared to *Budke et al.* (liver cancer). However, no specific DW has yet been developed for CE, so appropriate ones from diseases with similar morbidity have been used.

Cysticercosis was studied in three African countries. The highest zDALY on cysticercosis was calculated for Cameroon with data from 2008, followed by Tanzania (2012). However, Tanzania reported a higher ALE compared to Cameroon due to higher economic losses in the pig population. Mozambique data was only from the Agonia district; thus, the results are not comparable to the other countries. Although approximately only 0,9% of total zDALYs account for ALE in Cameroon, 2% in Mozambique, and 11% in Tanzania, respectively. When considering the zDALY per capita, Cameroon has the highest zDALY per capita (12 zDALYs per 1000 population), followed by Mozambique (6 zDALYs per 1000 population), and Tanzania (1 zDALY per 1000 population). Cameroon's cysticercosis estimated by *Praet et al.* was higher than the GBD's. For cysticercosis in Tanzania, Trevisan's estimation was also higher than GBD's, being the least difference in 2017 (the year of publication), around 24,166 DALYs. We assume the DALY on *T. solium* is higher than ALE, because it causes epilepsy in humans with high morbidity and mortality. Whereas the ALE on cysticercosis results only in organ condemnation. Furthermore, the lack of data on animals also contributes to a lower ALE. In Tanzania and Mozambique, pigs lose half of their value, while in Cameroon, the price usually is reduced by 30%. This demonstrates that cultural practices are relevant when estimating the impact or burden of a given condition on an animal population. It also shows that the zDALY metric is able to represent such differences effectively.

Generally, the impact of zoonoses is usually associated with low- and middle-income countries (LICs and LMICs). However, the studies in New Zealand and the Netherlands demonstrate that also high-income countries can suffer from losses in health, time, and money caused by zoonoses. Even though their impact is less than those in LICs and LMICs, they can worsen if appropriate preventive measures are not taken. For example, in the case of Q fever in the Netherlands, it was estimated that the loss of a culling milk goat is 100 times higher than a dose of the vaccine [25]. We estimated that in Netherlands Q-fever burden results to 2843 zDALYs, and only 2.86 is attributable to ALE. This could be because most of the infections due to *Coxiella burnetti* in animals are subclinical, and only result in abortions during late term. Furthermore, the control of Q-Fever is not included in these costs, however, authors mentioned that Q-fever control from the cost-utility perspective is expensive [25].

According to our findings, the burden of zoonoses impacts slightly more the human health sector, which is reflected in high DALYs rather than ALE, except for the estimations of the global cystic echinococcosis, leptospirosis in New Zealand, brucellosis in Kazakhstan, and zoonoses in Turkey (Fig 3). The total summed up estimates for our review resulted in 11,015,438 (95% CI: 6,235,971–15,806,100), with ALE representing almost half of the total zDALYs. However, it might be double counted for diseases such as rabies, and echinococcosis because estimates include both values for global burden and country specific burden.

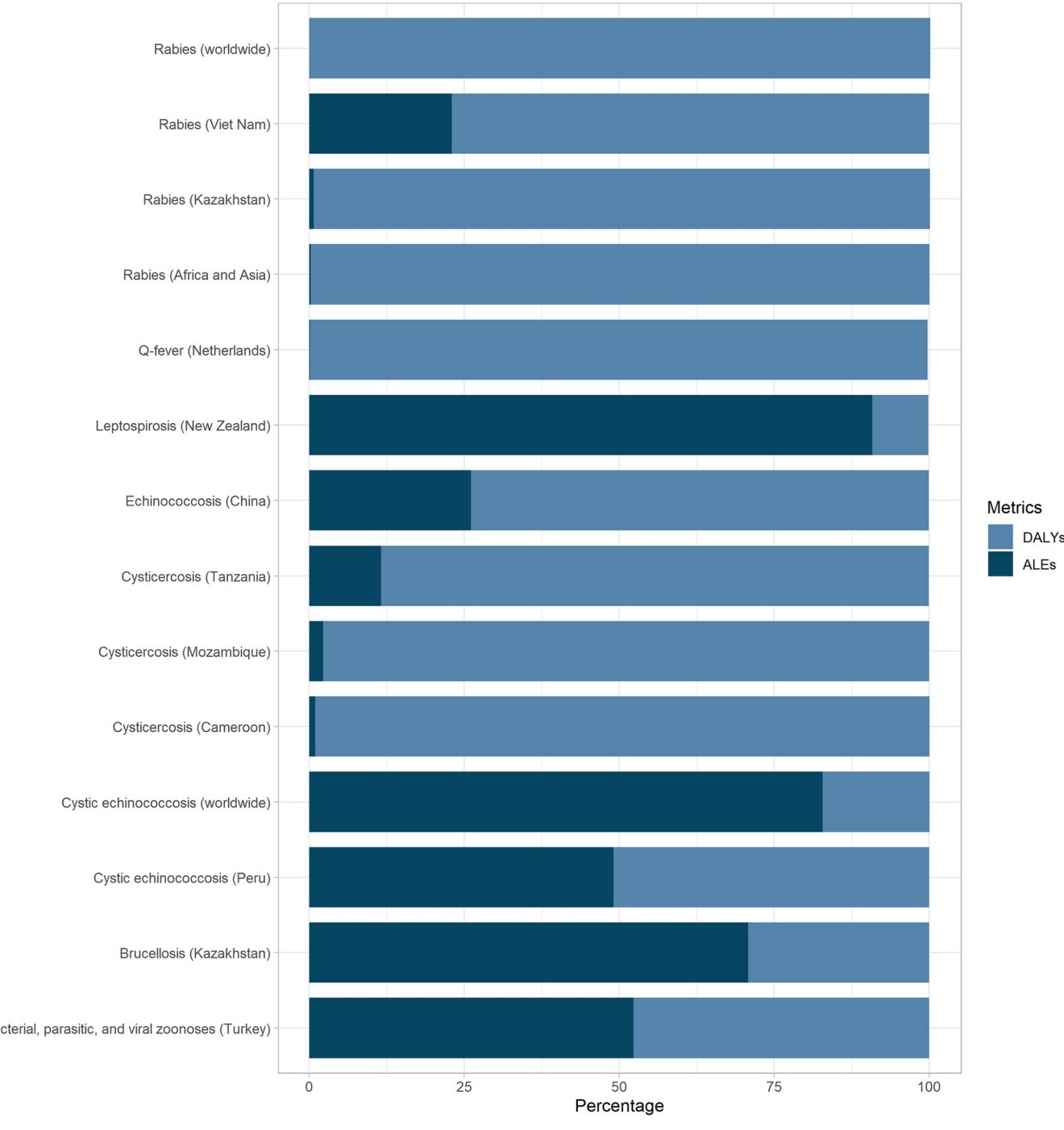

**Fig 3. Relative distribution of the DALYs and ALE among the studies.**

Excluded at the full-text screening stage, estimates provided by Roth et al. [39], when converted to animal health benefits saved, resulted in the same ballpark ratio of DALY to ALE as our estimations for Kazakhstan and Turkey.

The excluded studies with zDALYs were neurocysticercosis in Northern Lao PDR with 3497 zDALYs, cystic echinococcosis in Morocco 18,330 (95% CI 17,775–19,074), and

(bacterial, parasitic, viral, fungal) zoonoses in Paraguay with zDALYs of 62,178 (95% CI 48,696–77,188) (SI 3) [11–13]. The percentages attributable to animal burden were 0.4% for Northern Lao PDR, 99% for Morocco, and 69% for Paraguay. The studies with higher ALE than DALY demonstrate how the priorities of countries on zoonoses can change if animal populations are taken into consideration. When countries have higher DALYs compared to ALE, the first question one must ask is whether this is due to a lack of data from the animal population or if it is because only losses to farmers due to animal zoonosis account for the ALE.

Our estimations are based on the results of previous studies which is a limitation of this study, besides the small number of papers. In some cases, the data available for humans and animals were not from the same period, reducing the accuracy of the estimations (S3 Fig). Only three studies shared their code for the analysis (one of them partially), making the rest of the studies not reproducible. Also, the lack of availability of datasets following the FAIR principles did not allow us to obtain the confidence intervals of our choice. This shows the need for FAIR data application in the health area [40–42]. The lack of data continues to be a challenge, as the approach that is used to analyze it. We did not perform a meta-analysis due to the high variability among studies, including the type of study, and analysis design. This is also evidence of a lack of standardized methods to unify the burden caused by zoonoses in humans and animals in the past, and the unfamiliarity of the existing metrics available for that aim.

The strength of this study consists of an extensive literature search in different databases without an initial time restriction. Considering that the GBD study does not include most of the zoonoses burden, as well as the animal burden of zoonosis, we integrated this data into the human burden among the studies available worldwide. The DALY is a metric used to prioritize international disease-control investments. However, its use has been debated for various, primarily ethical, reasons. Among which is a limited applicability to neglected tropical diseases (NTDs). Most NTDs in this study have a low chronic morbidity that accounts only for a small portion of DALY. In low-income settings, where poverty is dominant, this low morbidity raises little attention. Half of the world hungry are subsistence farmers and rely heavily on agriculture for their livelihoods [43]. However, subsistence farming and hard physical work are common in those settings and the disabling effects of the NTDs are a main source of poverty. This circular causality cannot be captured through DALY calculations. The zDALY, at least, allows to include the burden from animal health losses, which are highly relevant in most poverty settings. How much subsistence farmers lose due to a zoonotic disease and how long it will take them to recover their losses should receive more attention in public health policy as it addresses an important determinant of human health which mainly consists of the social and economic environment [44].

Regarding vector-borne zoonoses, the only reported were tularemia and Crimean-Congo hemorrhagic fever (CCHF) in Turkey but without a direct association of their animal losses. We suggest establishing databases that incorporate human and animal diseases for each country, thus on a global scale. For example, complement the GBD database with ALEs to move towards better integration of human and animal health policies.

A remaining challenge for the zDALY are animals without traded economic value. Therefore, other methods for estimating the ALE component of the zDALY (e.g., willingness to pay, pairwise comparisons or direct time trade off) in analogy to ecosystem services should be explored [45]. Not only are more comprehensive metrics needed, but also a more integrative effort and support to face zoonosis in LICs and LMIC. For this endeavor, we consider the zDALY represents a step towards progress in zoonosis prioritization.

## Supporting information

**S1 Table. List of used terms for each electronic search.**
(DOCX)

**S2 Table. List of papers excluded at the full-text screening, with reasons of exclusion.**
(DOCX)

**S3 Table. List of countries included in the rabies studies (at global and continental levels).**
(DOCX)

**S4 Table. List of papers with zDALY estimates excluded from this systematic review.**
(DOCX)

**S5 Table. ROBIS tool.**
(DOCX)

**S6 Table. PRISMA Checklist.**
(DOCX)

**S1 Text. List of included studies.**
(DOCX)

**S1 Fig. Proportion of dual burden studies according to their zoonotic etiology: only one study included bacterial, viral, and parasitic zoonoses (labeled as: "All of them").**
(TIFF)

**S2 Fig. Publications of dual burden of zoonoses per range of years.**
(TIFF)

**S3 Fig. Gap between human data and year of publication: due to the use of secondary data, certain studies included old data for their analysis.** The same happened with human and animal data.
(TIFF)

## Acknowledgments

We thank Sabine Klein, the medical librarian, for assisting in the scientific publications search.

## Author Contributions

**Conceptualization:** Liz P. Noguera Z., Duriya Charypkhan, Sonja Hartnack, Paul R. Torgerson, Simon R. Rüegg.

**Data curation:** Liz P. Noguera Z., Duriya Charypkhan.

**Formal analysis:** Liz P. Noguera Z., Duriya Charypkhan.

**Funding acquisition:** Liz P. Noguera Z.

**Investigation:** Liz P. Noguera Z., Duriya Charypkhan.

**Methodology:** Liz P. Noguera Z., Duriya Charypkhan, Sonja Hartnack, Paul R. Torgerson, Simon R. Rüegg.

**Project administration:** Liz P. Noguera Z., Duriya Charypkhan.

**Resources:** Paul R. Torgerson.

**Software:** Liz P. Noguera Z.

**Supervision:** Sonja Hartnack, Paul R. Torgerson, Simon R. Rüegg.

**Validation:** Liz P. Noguera Z., Duriya Charypkhan, Sonja Hartnack, Paul R. Torgerson, Simon R. Rüegg.

**Visualization:** Liz P. Noguera Z.

**Writing – original draft:** Liz P. Noguera Z., Duriya Charypkhan.

**Writing – review & editing:** Liz P. Noguera Z., Duriya Charypkhan, Sonja Hartnack, Paul R. Torgerson, Simon R. Rüegg.

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
