## [Decision Letter · Decision Letter 0]

12 Sep 2022

Dear Mrs Noguera Zayas,

Thank you very much for submitting your manuscript "The dual burden of animal and human zoonoses: a systematic review" for consideration at PLOS Neglected Tropical Diseases. As with all papers reviewed by the journal, your manuscript was reviewed by members of the editorial board and by several independent reviewers. In light of the reviews (below this email), we would like to invite the resubmission of a significantly-revised version that takes into account the reviewers' comments. 

Your manuscript was evaluated by 3 reviewers and all required revisions prior to being acceptable for publication. Please review and respond to all the reviewer comments. The two most significant revisions to address are:

1) Clarification of some of the sampling and statistical techniques you used.

2) Figure 2 is of very poor quality and essentially unreadable - please replace this with a figure of higher quality for your re-submission.

There are also a number of items that require correction and editing (for example, Viet Nam vs Vietnam for consistency). 

We cannot make any decision about publication until we have seen the revised manuscript and your response to the reviewers' comments. Your revised manuscript is also likely to be sent to reviewers for further evaluation.

Sincerely,

Richard A. Bowen

Academic Editor

Victoria Brookes

Section Editor

Your manuscript was evaluated by 3 reviewers and all thought it was generally a valuable contribution, but required some revisions prior to being acceptable for publication. Please review and respond to all the reviewer comments. Your manuscript has been judged to require MAJOR REVISIONS. The two most significant revisions to address are:

1) Clarification of some of the sampling and statistical techniques you used.

2) Figure 2 is of very poor quality and essentially unreadable - please replace this with a figure of higher quality for your re-submission.

There are also a number of small and easy to correct items that require editing (for example, Viet Nam vs Vietnam for consistency).

We look forward to evaluating a revised version of this manuscript.

Reviewer's Responses to Questions

**Key Review Criteria Required for Acceptance?**

**Methods**

-Are the objectives of the study clearly articulated with a clear testable hypothesis stated?

-Is the study design appropriate to address the stated objectives?

-Is the population clearly described and appropriate for the hypothesis being tested?

-Is the sample size sufficient to ensure adequate power to address the hypothesis being tested?

-Were correct statistical analysis used to support conclusions?

-Are there concerns about ethical or regulatory requirements being met?

Reviewer #1: (No Response)

Reviewer #2: Yes.

Reviewer #3: -Are the objectives of the study clearly articulated with a clear testable hypothesis stated?

Yes

-Is the study design appropriate to address the stated objectives?

Yes

-Is the population clearly described and appropriate for the hypothesis being tested?

Yes

-Were correct statistical analysis used to support conclusions?

No. I think statistical methods needs more clarification.

-Are there concerns about ethical or regulatory requirements being met?

No

**Results**

-Does the analysis presented match the analysis plan?

-Are the results clearly and completely presented?

-Are the figures (Tables, Images) of sufficient quality for clarity?

Reviewer #1: (No Response)

Reviewer #2: Yes.

Reviewer #3: -Does the analysis presented match the analysis plan?

No. the used methods for pooling data is not clear. heterogenicity methods and results of them is not reported. 

-Are the results clearly and completely presented?

Tables are complex with some undefined headers.

-Are the figures (Tables, Images) of sufficient quality for clarity?

The quality of figures was too low and I was unable too assess them.

**Conclusions**

-Are the conclusions supported by the data presented?

-Are the limitations of analysis clearly described?

-Do the authors discuss how these data can be helpful to advance our understanding of the topic under study?

-Is public health relevance addressed?

Reviewer #1: (No Response)

Reviewer #2: Yes.

Reviewer #3: Are the conclusions supported by the data presented?

yes 

-Are the limitations of analysis clearly described?

yes

-Do the authors discuss how these data can be helpful to advance our understanding of the topic under study?

yes

**Editorial and Data Presentation Modifications?**

Reviewer #1: (No Response)

Reviewer #2: I suggest improving the quality of the figures, especially figure 2.

Reviewer #3: Dear editor

Hi

The topic of this study is attractive and I think it would be improve using reviewers comment. I recommended major revision to revise the statistical methods and results.

Best

**Summary and General Comments**

Reviewer #1: This study presents the first systematic review that estimates the dual burden of zoonoses in humans and domestic animals based on studies available worldwide. This is an interesting study, which provides important information for the prioritization of the control and prevention of zoonotic diseases. Please see below my comments and suggestions.

ABSTRACT 

Lines 25-27: I suggest rephrasing this sentence as it is currently unclear whether the ALE are for rabies and echinococcosis only. 

INTRODUCTION 

Line 63: I suggest adding a definition of Disability Adjusted Life Years (DALY). 

METHODS 

Line 91: Did you include studies written in a language other than English?

I am concerned that your search strategy missed relevant studies: 

- Did you consider doing a reference list search of studies included for full-text review or in the review for potential additional studies? 

- To what extent do you think using the term “zoonoses” rather than the actual zoonotic disease names in your literature search (i.e., rabies, echinococcosis, cysticercosis, brucellosis, leptospirosis, etc.), decreased the number of studies identified? 

RESULTS 

Line 160: Replace “e.g.” with “for example”.

DISCUSSION

While it is clear why the three studies already including zDALY estimates were excluded from the actual review, it would have been relevant to include them in the discussion section, and discuss their results more extensively (e.g., include them in an overall total summed up estimate of zDALY, etc.). 

Line 293: You mention “the lack of availability of datasets following the FAIR principles did not allow us to obtain the confidence intervals of our choice”; did you try contacting the corresponding authors of these studies to obtain the data? Same question for the studies that did not share their code. 

Line 314: Which determinant of human health?

OTHER

The use of “Vietnam” or “Viet Nam” should be consistent throughout the manuscript.

Please check the consistency in the references – sometimes they are cited after the sentence (e.g., “.[8]” instead of “[8].”).

Abbreviations should be defined only once at first use, and not multiple times throughout the manuscript (e.g., ALE and zDALY). Additionally, all abbreviations should be defined (e.g., LIC and LMIC are not).

Reviewer #2: The article presents a systematic review that addresses an innovative theme within the concept of One Health. The review was carried out with adequate methodological rigor.

Reviewer #3: Dear author 

thanks for your great study. I think you chose great topic, but methods and results section needs more clarification; you didn't explain well about statistical methods (pooling the results method, heterogenicity assessment method, ...) and I think tables are complex, please see my comments in you article PDF.

your claim in PRISMA checklist was not according pages of PDF.

PLOS authors have the option to publish the peer review history of their article (what does this mean?). If published, this will include your full peer review and any attached files.

Reviewer #1: No

Reviewer #2: No

Reviewer #3: No
---

## [Editor Report · Decision Letter 1]

2 Oct 2022

Dear Mrs Noguera Zayas,

We are pleased to inform you that your manuscript 'The dual burden of animal and human zoonoses: a systematic review' has been provisionally accepted for publication in PLOS Neglected Tropical Diseases.

Best regards,

Richard A. Bowen

Academic Editor

Victoria Brookes

Section Editor

---

## [Editor Report · Acceptance letter]

10 Oct 2022

Dear Mrs Noguera Z.,

We are delighted to inform you that your manuscript, "The dual burden of animal and human zoonoses: a systematic review," has been formally accepted for publication in PLOS Neglected Tropical Diseases.

Best regards,

Shaden Kamhawi

co-Editor-in-Chief

Paul Brindley

co-Editor-in-Chief
